# Visual recognition and prediction analysis of China's real estate index and stock trend based on CNN-LSTM algorithm optimized by neural networks

Ningyan Chen * 

Business School, University of Aberdeen, Aberdeen, The United Kingdom

* n.chen.21@abdn.ac.uk

## Abstract

Today, with the rapid growth of Internet technology, the changing trend of real estate finance has brought great an impact on the progress of the social economy. In order to explore the visual identification (VI) effect of Convolutional Neural Network and Long Short-Term Memory (CNN-LSTM) algorithm based on neural network optimization on China's real estate index and stock trend, in this study, artificial neural network (ANN) algorithm is introduced to predict its trend. Firstly, LSTM algorithm can effectively solve the problem of vanishing gradient, which is suitable for dealing with the problems related to time series. Secondly, CNN, with its unique fine-grained convolution operation, has significant advantages in classification problems. Finally, combining the LSTM algorithm with the CNN algorithm, and using the Bayesian Network (BN) layer as the transition layer for further optimization, the CNN-LSTM algorithm based on neural network optimization has been constructed for the VI and prediction model of real estate index and stock trend. Through the performance verification of the model, the results reveal that the CNN-LSTM optimization algorithm has a more accurate prediction effect, the prediction accuracy is 90.55%, and the prediction time is only 52.05s. At the same time, the significance advantage of CNN-LSTM algorithm is verified by statistical method, which can provide experimental reference for intelligent VI and prediction of trend of China real estate index and property company stocks.

## 1. Introduction

Since the establishment of the People's Republic of China, the real estate industry has experienced a long development process. Since the beginning of the 21st century, the price of real estate is increasing at an astonishing speed, and the real estate bubble is vulnerable to the impact of the subprime crisis. People borrow large amounts of money from banks to buy houses. Many individuals owe more than 60% of the total value of their homes. Banks sell loan certificates at relatively low prices to financial institutions, which package housing loans as financial products and sell them to investors [1–3]. Once the real estate bubble bursts, the real estate market collapses and housing prices plummet. Buyers and investors of financial products will face a crisis. During the subprime crisis, buyers took on above-market loans to buy

**Funding:** The author received no specific funding for this work.

**Competing interests:** The authors have declared that no competing interests exist.

homes. Many individuals defaulted on mortgages because they could not pay, banks could not collect loans, and capital chains were broken. Businesses could not borrow, and all types of businesses shrank [4]. Therefore, accurate prediction of the development trend of the real estate industry has become the focus of scholars in related fields.

The continuous optimization of AI algorithms has also promoted, intellectualized, and innovated real estate finance. In real estate, predicting financial stock returns is the most direct econometric or statistical method to analyze the market trend [5]. As one of the AI algorithms, Deep Learning (DL) can use historical prices, technical indicators, and other information to mine hidden change rules and patterns in the stock market or futures market. DL can predict stock prices (through regression) and stock trends (through classification) [6, 7]. Among many DL algorithms, the Convolutional Neural Network (CNN) can reduce the number of parameters. Meanwhile, it extracts useful features by connecting the convolutional layer with the previous layer through local connection and weight sharing [8]. A Recurrent Neural Network (RNN) establishes weight connections between neurons in the same layer. As a result, the hidden layer state of the last moment is included in calculating the current moment, and Long Short-Term Memory (LSTM) introduces the cell state. It solves the vanishing gradient in traditional RNN and effectively learns the relationship before and after the time series [9]. Therefore, fusing DL algorithm models with different advantages might optimize the performance of predicting the trend of the real estate industry.

To sum up, with the continuous improvement and optimization of AI algorithms, the real estate index and the trend of stock are influenced by many factors and are extremely open. Intelligent prediction of the real estate indexes and the stock trend is of great practical value to social and economic development. Thereupon, the innovation of this study is that aiming at the openness of the real estate finance field, while improving the RNN algorithm, it further introduces CNN with less application to optimize the algorithm model. Finally, the LSTM algorithm is combined with the CNN algorithm, and the BN layer is used as the transition layer for further optimization. The CNN-LSTM algorithm based on DL optimization is constructed to predict the real estate index and stock trend, thereby providing a reference for the subsequent intelligent forecast of the real estate index and stock trend and the structure optimization of property companies.

## 2. Literature review

### 2.1 Application status of DL algorithms in the financial field

Financial time series data often contain noise and non-stationary behavior in the stock and index samples. It is very challenging for the business community and researchers to model and forecast the real estate price. Many scholars in related fields have conducted research. Abualigah & Diabat (2021) analyzed Sine Cosine Algorithm (SCA), and verified SCA's performance on similar algorithms through a series of computational experiments [10]. Hou et al. (2021) proposed a hybrid model based on Graph Convolutional Network (GCN) and LSTM for predicting the stock market. This model was expected to be used in applications with hidden spatial correlation to improve time series prediction [11]. Althelaya et al. (2021) established a Deep Neural Network (DNN)-based stock price prediction model. They fused DL with multi-resolution analysis to enhance stock prediction accuracy. The model was proven effective over other models in evaluating the time series of the S&P 500 index [12]. Song et al. (2021) put forward a filling-based Fourier transform denoising algorithm. The algorithm eliminated the spectral-domain noisy waveform of monetary time-sequence data and ameliorated information differences on both sides when recovering to the initial time sequences. The authors forecasted the closing prices of the S&P 500 index, the Shanghai Composite Index, and the Korean

Kospi Index. Thus, integrating the DL model and the denoising technique improved the primary model's prediction performance and alleviated the time delay problem [13]. Kamara (2022) studied an end-to-end algorithm for stock price prediction by adding DL to the Attention Mechanism to extract stock data information. This scheme performed dramatically better than independent DL schemes, statistical algorithms, and other Machine Learning models in experiments [14]. Song et al. (2023) combined the DL method with the generalized autoregressive conditional heteroskedasticity and mixed data sampling (GARCH-MIDAS) model to predict the volatility of the stock market containing low-frequency macroeconomic variables to process mixed-frequency data. Compared with the existing model algorithm, the results have the best prediction effect [15].

## 2.2 The development status of real estate finance using intelligent algorithm optimization

Real estate is a complex industry with many influencing factors. It is of great significance to understand its development status for socio-economic development. Abualigah et al. (2021) proposed a new population-based optimization method. Through experimental analysis, it was found that this method can obtain the predicted optimal solution when applied to practical engineering [16]. Lv et al. (2021) argued that the constructed vertical market system was very stable from the perspective of the vertical market. Self-operated retailers had more advantages, providing an experimental reference for future Smart City construction and economic development [17]. Zhao et al. (2021) designed the Cumulative Prospect Theory-based Interactive and Multicriteria Decision Making (CPT-TODIM) method to obtain the attribute weight and improve the calculation's rationality. The discussed method was applied to stock investment selection, and the stock investment selection was demonstrated based on the proposed method [18]. Yang et al. (2022) applied the Particle Swarm Optimization (PSO) algorithm based on double-bottom mapping to cluster real estate-related data collected from public websites. Three basic variables of real estate prices were revealed: money supply, population, and rent. The finding could help the government manage the relationship between rent and the real estate market [19].

## 2.3 Summary

Based on the research and analysis of the above scholars, the DL algorithm has been widely applied in the financial field, and it is feasible for analyzing the relevant results. Mostly, RNN is used for modeling the financial time series. Research on using CNN to predict stocks is rare. Although many scholars try to predict the financial trend of indexes and stocks, some changes have taken place in the real estate industry under the influence of intelligent environment, so there is still a great unknown and research blank. Thereby, in this study, aiming at the openness of the real estate finance field, while improving the RNN algorithm, CNN with less application is introduced to further optimize the algorithm model. Through an exploration of the prediction of indexes and stocks in the real estate field, different experiences are provided for the subsequent trend and development of the real estate field.

## 3. Use the CNN-LSTM algorithm optimized by DL to predict, evaluate, and analyze the real estate index and stock trend

### 3.1 Real estate index and stock data preprocessing analysis

As a key component of the social economy, the real estate industry has a crucial economic value for predicting index and stock trends. The predicting model's prediction accuracy and

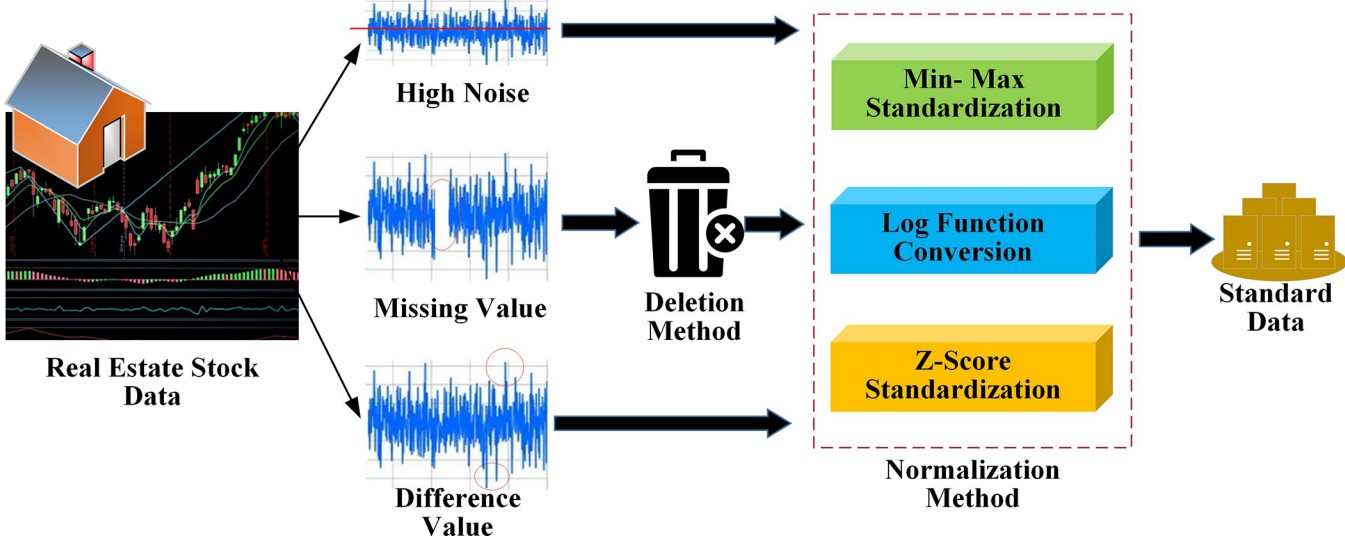

**Fig 1. Real estate index and stock data preprocessing.**

training speed are often affected by the high noise, missing value, and inconsistency in the original real estate index and stock data. Therefore, the financial time series must be preprocessed first, as in Fig 1. The missing values can be preprocessed by deleting or interpolation [20, 21]. The stock data are volatile and noisy. The interpolation method can fill the missing data. However, there might be a large gap between the filled value and the real value, thus affecting the model accuracy. Accordingly, this study chooses to delete the missing values.

The original data's closing price and trading volume show that the order of magnitude of different characteristic data is quite different. Thus, the data must be normalized. All indicators are processed in the same order of magnitude, and the data are transformed into values between specified intervals. Finally, the influence of outliers and extreme values are avoided indirectly through centralization, which speeds up the convergence speed and improves the accuracy of the optimal solution found. Normalization methods cover min-max, log function transformation, and Z-Score [22].

First, min-max standardization. The output value is mapped between [0, 1] by linearly changing the original data. When new data is added, max and min may change and need to be calculated again. The conversion function of data $x$ is illustrated in Eq (1):

$$\tilde{x} = \frac{x - \min}{\max - \min} \tag{1}$$

Here, $\tilde{x}$ refers to the normalized result of x, min, and max represent the minimum and maximum values of the sample data, respectively.

Second, log function conversion. Data are converted by log function with base 10 by Eq (2):

$$\tilde{x} = \frac{\log_{10}(x)}{\log_{10}(\max)} \tag{2}$$

In Eq (2), max stands for the maximum value of sample data. All data must be $\geq 1$ before log function conversion.

Third, Z-Score normalization. The mean value of the processed index and stock data is 0, and the standard deviation is 1. In this way, the data are normalized to standard distributed

data. The transformation function is as follows:

$$\tilde{x} = \frac{x - \mu}{\sigma} \tag{3}$$

In Eq (3), $\mu$ is the mean of the sample, $\sigma$ denotes the standard deviation of the sample.

In this study, the input characteristics of all real estate indexes and stocks are normalized using the Z-Score method, and the mean and variance are taken as the normalization parameters. The normalization on returns is as follows:

$$\tilde{r}_t^s = \frac{r_t^s - \mu^s}{\sigma^s} \tag{4}$$

$\tilde{r}_t^s$ refers to the normalized result of the return $r_t^s$.

### 3.2 DL algorithm and its improvement analysis

As one of the DL algorithms, CNN is a feature extractor with excellent performance and a high degree of automation. The trend of the real estate index and stock can be predicted and recognized by CNN. The end-to-end method eliminates the process of manual feature extraction and excessively extract its interior semantic characteristics [23, 24]. The basic flow of CNN is drawn in Fig 2.

In Fig 2, the CNN-based real estate index and stock prediction model covers a convolutional layer, pooling layer, and fully connected (FC) layer. As the most vital hierarchical structure, the convolutional layer can extract feature information from input image data. Multiple feature maps representing image data can be obtained through feature extraction of the convolutional layer. Each characteristic graph indicates a kind of internal semantic characteristic data of image data [25]. The following are the structure parameter settings. ① Input feature mapping group. Each slice matrix $X^d \in \square^{H \times W}$ in the 3D tensor $X \in \square^{H \times W \times D}$ is an input feature

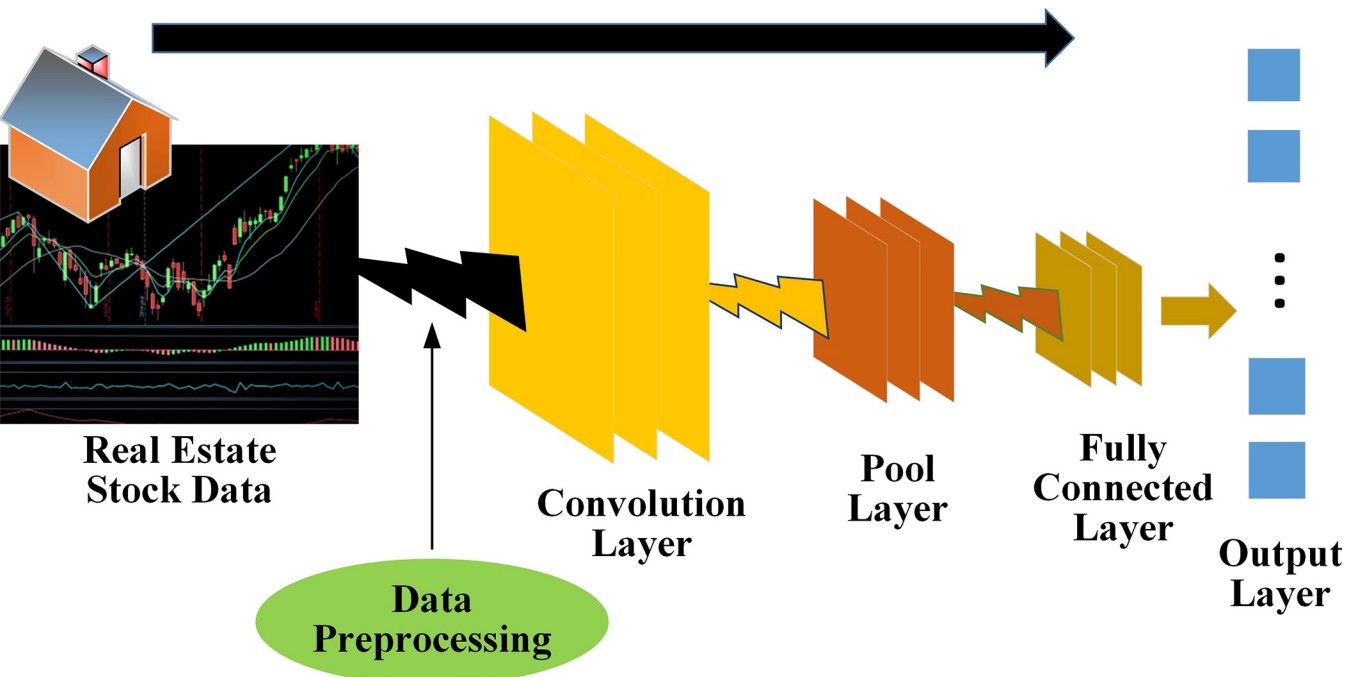

**Fig 2. Basic flow of CNN applied to real estate index and stock.**

mapping, $1 \leq d \leq D$. ② Output feature mapping group. Each slice matrix $Y^P \in \square^{H' \times W'}$ in the 3D tensor $Y \in \square^{H' \times W' \times P}$ is an output feature mapping, $1 \leq p \leq P$. ③ Convolution kernel (CK). All slice matrices in the four-dimensional tensor are two-dimensional CKs, $1 \leq d \leq D$, and $1 \leq p \leq P$. Among the above three design parameters, multiple convolution filters $\{W^{p,1}, W^{p,2}, \cdots, W^{p,D} : W \in \square^{h \times w \times D \times P}\}$ are applied to the input feature mapping group $\{X^1, X^2, \cdots, X^D : X \in \square^{H \times W \times D}\}$ for calculation. The characteristic mapping results produced by convolution filters are summed up. Then the output of the convolution layer is obtained by adding the bias $b$, as follows:

$$Z^p = W^p \otimes X + b^p = \sum_{d=1}^{D} W^{p,d} \otimes X^d + b^p \qquad (5)$$

In Eq (5), $W \in \square^{h \times w \times D \times P}$ is often a channel CK. Then, the subsequent output feature mapping is obtained by calculating the nonlinear activation function (NAF) by Eq (6):

$$Y^p = f(Z^p) \qquad (6)$$

Here, $f(.)$ is the NAF.

The pooling layer, as known as the sub-collection layer, is responsible for selecting and reducing gained characteristic vectors to streamline the parameters. Commonly used pooling layers are maximum pooling (MaP) and mean pooling (MeP). The MaP layer extracts the maximum of neuron nodes in the regional receptive field by Eq (7):

$$Y_{m,n}^d = \max_{i \in R_{m,n}^d} x_i \qquad (7)$$

Here, $x_i$ refers to the activation value of each neuron in region $R$. MeP calculates the average value of all neural nodes in the local receptive field by Eq (8):

$$Y_{m,n}^d = \frac{1}{|R_{m,n}^d|} \sum_{i \in R_{m,n}^d} x_i \qquad (8)$$

The $M' \times N'$ areas of every input characteristic graph $X^d$ are sampled to gain the characteristic graph of the output of the pooling layer $Y^d = \{Y_{m,n}^d\}$, $1 \leq h \leq H$, $1 \leq w \leq W'$. Reducing neural nodes in the lower pooling layer can make the CNN model rather robust. Local significant characteristics of image data are extracted so that the network has partial invariant characteristics to image data. Additionally, dimensionality reduction is performed on the output characteristic mapping collection to eliminate model overfitting due to redundant features.

In a FC layer, each unit in the previous layer is connected to all units in the following layer. Local features are extracted through convolution operation to stitch all the extracted local features into a complete data feature map. In this process, various weights need to be assigned to different features to form the weight matrix (WM). It mainly acts as a "classifier."

In CNN's error propagation, the error $z$ will be obtained by batch feedforward of $n$ randomly selected samples. The loss function of the $l$ layer can be obtained by Eq (9) and Eq (10).

$$\frac{\partial z}{\partial \omega^l} = 0 \qquad (9)$$

$$\frac{\partial z}{\partial x^l} = x^l - y \qquad (10)$$

Therefore, there will be two parts of the derivative process for each layer of operation: the

derivative of the error with respect to the parameter of the $l$ layer $\frac{\partial z}{\partial \omega^l}$ and the derivative of the error with respect to the input of the $l$ layer $\frac{\partial z}{\partial x^l}$. The parameter update $\frac{\partial z}{\partial \omega^l}$ is obtained $\omega^l$ by taking the derivative of the error with respect to the parameter by Eq (11):

$$\omega^l \leftarrow \omega^l - \eta \frac{\partial z}{\partial \omega^l} \qquad (11)$$

In Eq (11), $\eta$ refers to the step size of each gradient descent, usually inversely proportional to the number of training iterations.

The error $x^l$ propagates back and forth from the derivative $\frac{\partial z}{\partial x^l}$, which can be regarded as the error signal of the final error transmitted to the $l$ layer. For example, when the error update derivative is transmitted to the $l$ layer, the error derivative of the $l+1$ layer is $\frac{\partial z}{\partial x^{l+1}}$. The updated value of the $l$ layer parameters can be obtained by calculating the value of $\frac{\partial z}{\partial \omega^l}$ and $\frac{\partial z}{\partial x^l}$. According to the chain rule, Eqs (12) and (13) can be obtained:

$$\frac{\partial z}{\partial (vec(\omega^l)^T)} = \frac{\partial z}{\partial (vec(x^{l+1})^T)} \cdot \frac{\partial (vec(x^{l+1}))}{\partial (vec(\omega^l)^T)} \qquad (12)$$

$$\frac{\partial z}{\partial (vec(x^l)^T)} = \frac{\partial z}{\partial (vec(x^{l+1})^T)} \cdot \frac{\partial (vec(x^{l+1}))}{\partial (vec(x^l)^T)} \qquad (13)$$

Thus, $\frac{\partial z}{\partial x^{l+1}}$ can be calculated at $l+1$ layer and needs to be vectorized and transposed to be obtained $\frac{\partial z}{\partial (vec(x^{l+1})^T)}$. This calculation is convenient for parameter updates. Then, the parameters of this layer are updated according to Eqs (12) and (13), and the error $\frac{\partial z}{\partial x^l}$ of this layer is passed to the previous layer. This cycle is repeated until the parameters of the first layer are updated until a batch parameter update is completed.

RNN, as one of the DL algorithms, is mainly employed to process time series data. Calculating the connection between distant nodes involves multiple multiplications of the Jacobian matrix. The RNN-based model can lead to gradient disappearance or gradient inflation. LSTM is a special RNN model proposed to solve the RNN's gradient dispersion. In the traditional RNN, Backpropagation Through Time (BPTT) is used in the training algorithm. When the time is relatively long, the residual to be returned will decrease exponentially, resulting in a slow update of network weights. At this time, RNN fails to present a long-term memory. Thereby, a memory unit is needed to store memory, thus the LSTM model [26, 27]. Fig 3 compares the LSTM and RNN.

Apparently, the biggest difference between RNN and LSTM is that the top layer of LSTM has an information conveyor belt named "cell state," which is where the information is stored. The forgetting gate ($f$), input gate ($i$), output gate ($o$), and a memory unit ($c$) are introduced into the hidden layer of LSTM for information memory, update, and utilization. Cell state and hidden state are adopted to store internal state. The forgetting gate is employed to decide which information needs to be removed from the cell state, that is, "forgetting"; The input gate is used to select the essential input data to be saved in the cell state, and the state gate further determines the update approach of the cell state of LSTM based on the input gate. Introducing message controlled by the forgetting gate can produce additional memory content $C_t$. The output gate (OG) is used to determine what information needs to be output $h_t$, which will be selected in the new memory output decision. The calculations involved in LSTM are exhibited

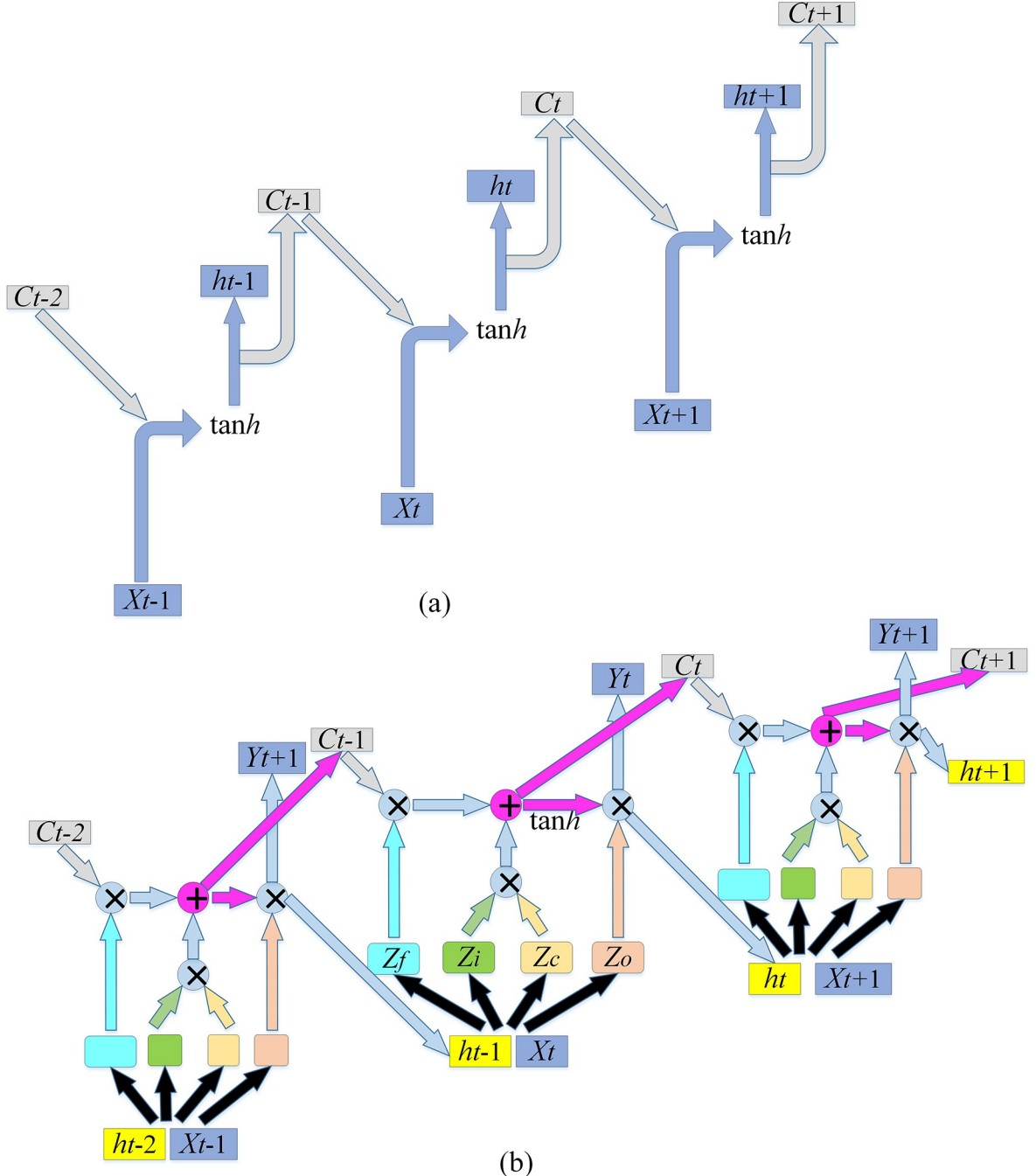

**Fig 3.** Comparison of RNN and LSTM structures (a. RNN; b. LSTM).

in Eqs (14)–(19):

$$f_t = \sigma(W_f \cdot [h_{t-1}, x_t] + b_f) \tag{14}$$

$$i_t = \sigma(W_i \cdot [h_{t-1}, x_t] + b_i) \tag{15}$$

$$\tilde{C}_t = \tanh(W_C \cdot [h_{t-1}, x_t] + b_C) \tag{16}$$

$$C_t = f_t * C_{t-1} + i_t * \tilde{C}_t \tag{17}$$

$$o_t = \sigma(W_o \cdot [h_{t-1}, x_t] + b_o) \tag{18}$$

$$h_t = o_t * \tanh(C_t) \tag{19}$$

In Eqs (14) to (19), $W_f$ refers to the WM of the OG, $x_t$ is the current input, $h_t$ represents the output of the previous step, and $b$ denote the bias term. $W_i$ means the WM of the OG, $W_C$ stand for the WM generated by the cell state. $W_o$ signifies the WM of the OG.

LSTM is a threshold RNN. In other words, the weight of the self-loop is changed by increasing the input threshold, forgetting threshold, and output threshold. In this way, the integration scale at different moments can be dynamically changed while the model parameters are fixed. Thereby, it avoids gradient disappearance or gradient expansion.

The LSTM algorithm is further optimized to predict the stock with a two-layer LSTM, as signified in Fig 4.

## 3.3 Construction and analysis of real estate index and stock trend prediction models using CNN-LSTM algorithm optimized by DL

This section introduces a DL algorithm to predict the real estate index and stock trend. LSTM algorithm can solve the gradient disappearance problem and can lend to processing financial time series data well. Meanwhile, with its unique fine-grained convolution operation, CNN can extract effective feature information from the original input, which has significant advantages in classification problems. Therefore, the advantages of LSTM and CNN are fused. The BN is introduced in the fusion model as the transition layer. The specific CNN-LSTM algorithm optimized by DL for real estate index and stock trend prediction is demonstrated in Fig 5.

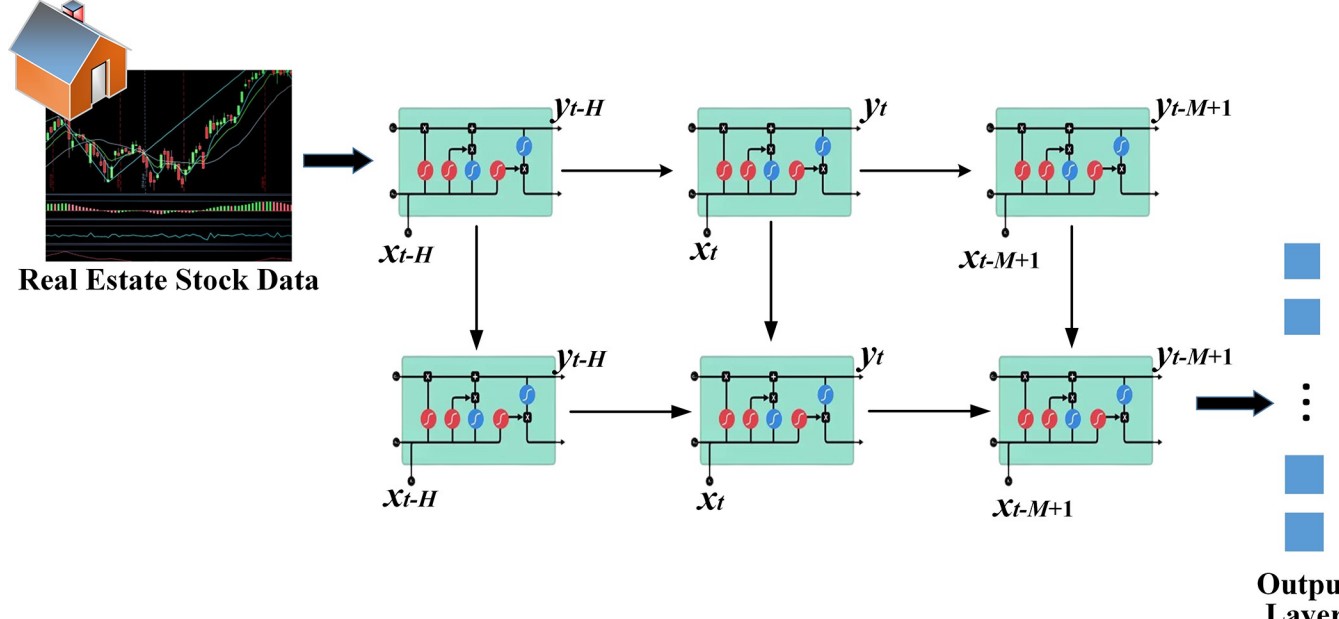

**Fig 4. Schematic diagram of LSTM network applied to real estate index and stock prediction.**

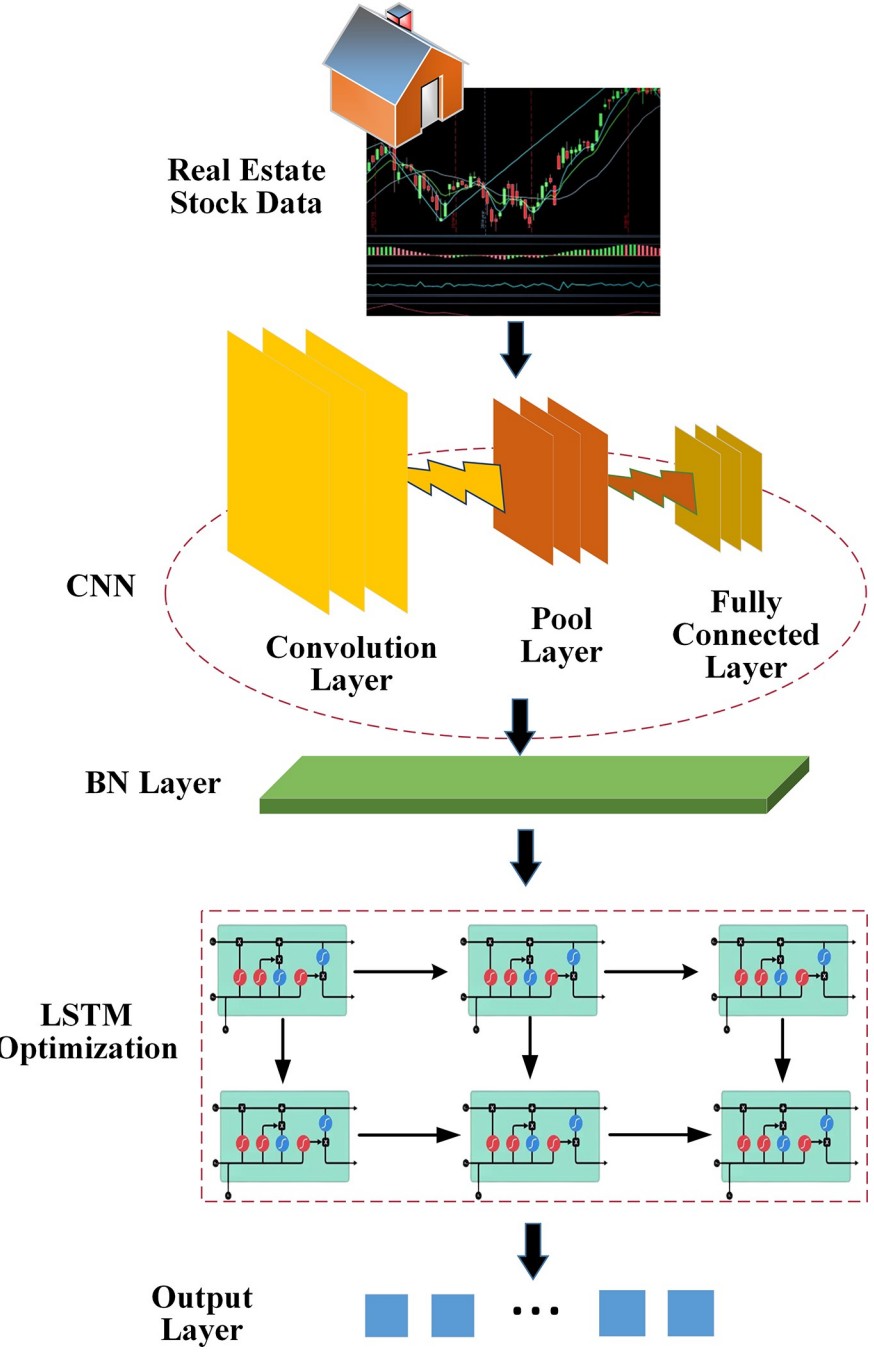

**Fig 5. The prediction model framework of real estate index and stock trend based on CNN-LSTM algorithm optimized by DL.**

In Fig 5, a cycle's real estate index and stock trend data are first converted into 2D image data, and then features are extracted through CNN. Specifically, the inventory sequence is defined as:

$$X_{r \times m} = [x_1, x_2, \ldots, x_{r-1}, x_r] \tag{20}$$

Here, each $x$ is an m-dimensional vector, so the result is $r$ times the matrix form of m. Therefore, the features of this matrix can be extracted through CNN. Then, 64 filters are used for feature extraction and then activated with the ReLu function. Then Max pooling is used for pooling, and Dropout with a probability of 0.3 is added to prevent overfitting. Finally, a sequence is an output as input to the subsequent LSTM. Finally, financial time series are modeled by optimizing the two-layer LSTM algorithm to extract feature maps.

In the real estate index and stock trend prediction, the volume and price characteristics are composed of stock volume and price information. The correlation return characteristics are composed of multiple stock returns with high correlation.

The volume-price feature uses information such as opening price, closing price, and stock trading volume to make forecasts. The experiment uses the data of the first 30 days to predict the price trend of the next day. The daily volume and price characteristics of each stock $feature_t^s$ are expressed by Eqs (21) and (22):

$$feature_t^s = [price_{t-29}^s, price_{t-28}^s, \cdots, price_t^s] \tag{21}$$

$$price_t^s = [open_t, high_t, close_t, low_t, volumn_t, amount_t] \tag{22}$$

Here, $price_t^s$ refers to the volume and price characteristics of the real estate index and stock $s$ on the day $t$, including the opening price $open_t$, highest price $high_t$, closing price $close_t$, lowest price $low_t$, trading volume $volumn_t$ and trading volume $amount_t$ of the stock in turn.

Related return characteristics can affect other related real estate indexes and the future price of stocks. Therefore, K highly correlated real estate indexes and stock returns are used to construct characteristics for these stocks:

$$feature_t = [return_{t-29}, return_{t-28}, \cdots, return_t] \tag{23}$$

$$return_t = [\tilde{r}_t^1, \tilde{r}_t^2, \cdots, \tilde{r}_t^R] \tag{24}$$

$return_t$ means the return list of K stocks on day $t$.

### 3.4 Simulation and simulation analysis

The purpose is to verify the performance of the proposed CNN-LSTM algorithm optimized by DL in predicting real estate indexes and stock. Matlab is used for simulation. The date, opening, high, close, low, and turnover of Sohu Securities China Real Estate Index (399241), Poly Development (600048), Gedi Group (600383), and Nanshan Holdings (002314) are collected by tushare for the experiment. The data cover November 12, 2019, to May 12, 2022.

Firstly, the model reported here is compared the CNN, LSTM, and the proposed CNN-LSTM optimized by DL in predicting the closing price of real estate indexes and stocks to verify its universality. Table 1 lists the parameter configuration of CNN, LSTM, and CNN-LSTM.

The statistical methods of ANOVA and T-test are applied to compare the algorithms, so as to determine whether there are major differences between the proposed algorithm and other comparison algorithms [28]. For this test, the statistical hypothesis can be expressed as follows. Null hypothesis ($H_0$) indicates that the difference between the two groups is not significant. Alternative hypothesis ($H_1$) indicates that there is an obvious difference between the means of the two groups of the population, that is, there is a difference. Furthermore, the CNN, LSTM, and the proposed CNN-LSTM optimization algorithm are compared and analyzed with the prediction accuracy and time required by the model algorithm proposed by Kamara (2022) and Song et al. (2023) scholars in related fields.

**Table 1. Parameter configuration of the CNN, LSTM, and CNN-LSTM algorithm.**

|  | Name | Output dimension value |
|---|---|---|
| CNN | input | (batch, 30, 6, 1) |
|  | conv1 | (batch, 30, 6, 32) |
|  | conv2 | (batch, 30, 6, 64) |
|  | pool | (batch, 15, 3, 64) |
|  | dropout | (batch, 15, 3, 64) |
|  | dense | (batch, 64) |
|  | output | (batch, 2) |
| LSTM | input | (batch, 30, 6) |
|  | lstm1 | (batch, 30, 32) |
|  | lstm2 | (batch, 64) |
|  | output | (batch, 2) |
| CNN-LSTM | input | (batch, 30, 6) |
|  | lstm | (batch, 30, 32) |
|  | reshape | (batch, 30, 32, 1) |
|  | batch_normalization (BN) | (batch, 30, 32, 1) |
|  | conv | (batch, 30, 32, 32) |
|  | pool | (batch, 15, 16, 32) |
|  | dropout | (batch, 7680) |
|  | dense | (batch, 32) |
|  | output | (batch, 2) |

The proposed CNN-LSTM model's neural network hyperparameters are set. The number of iterations is 80, the simulation time is 2,000s, and the Batch Size is 128. At the same time, hardware and software are configured with a Linux 64bit operating system, Python 3.6.1, and the PyCharm development platform; Intel core i7-7700@4. 2GHz 8 Cores CPU, Kingston DDR4 2400MHz 16G RAM, and Nvidia GeForce 1060 8G GPU.

## 4. Results and discussions

### 4.1 Comparative analysis of prediction performance of each model algorithm

The CNN, LSTM, and the proposed CNN-LSTM algorithm are used to predict the closing price of China Real Estate Index (399241), Poly Development (600048), Gedi Group (600383), and Nanshan Holdings (002314). Figs 6–8 plot and compare the results.

Each real estate index and stock is forecasted by using CNN algorithm in Fig 6. Evidently, the trend in predicting real estate indexes and stocks of China Real Estate Index (399241), Poly Development (600048), Gemdale Group (600383), and Nanshan Holdings (002314) is consistent with the actual value. However, the accuracy of predicting the maximum and the minimum values is still low.

The real estate index and stock are predicted using the LSTM algorithm in Fig 7. The performance of LSTM is different in different stocks. In particular, the prediction of the China real estate index (399241) is most consistent with the actual value. This may be because the China Real Estate index (399241) is composed of a combination of stocks in related fields, which is compatible with other stocks. The trend of Poly Development (600048), Gemdale Group (600383), and Nanshan Holdings (002314) can be clearly predicted. However, its prediction accuracy is not as high as that of the China Real Estate Index (399241). Thus, with

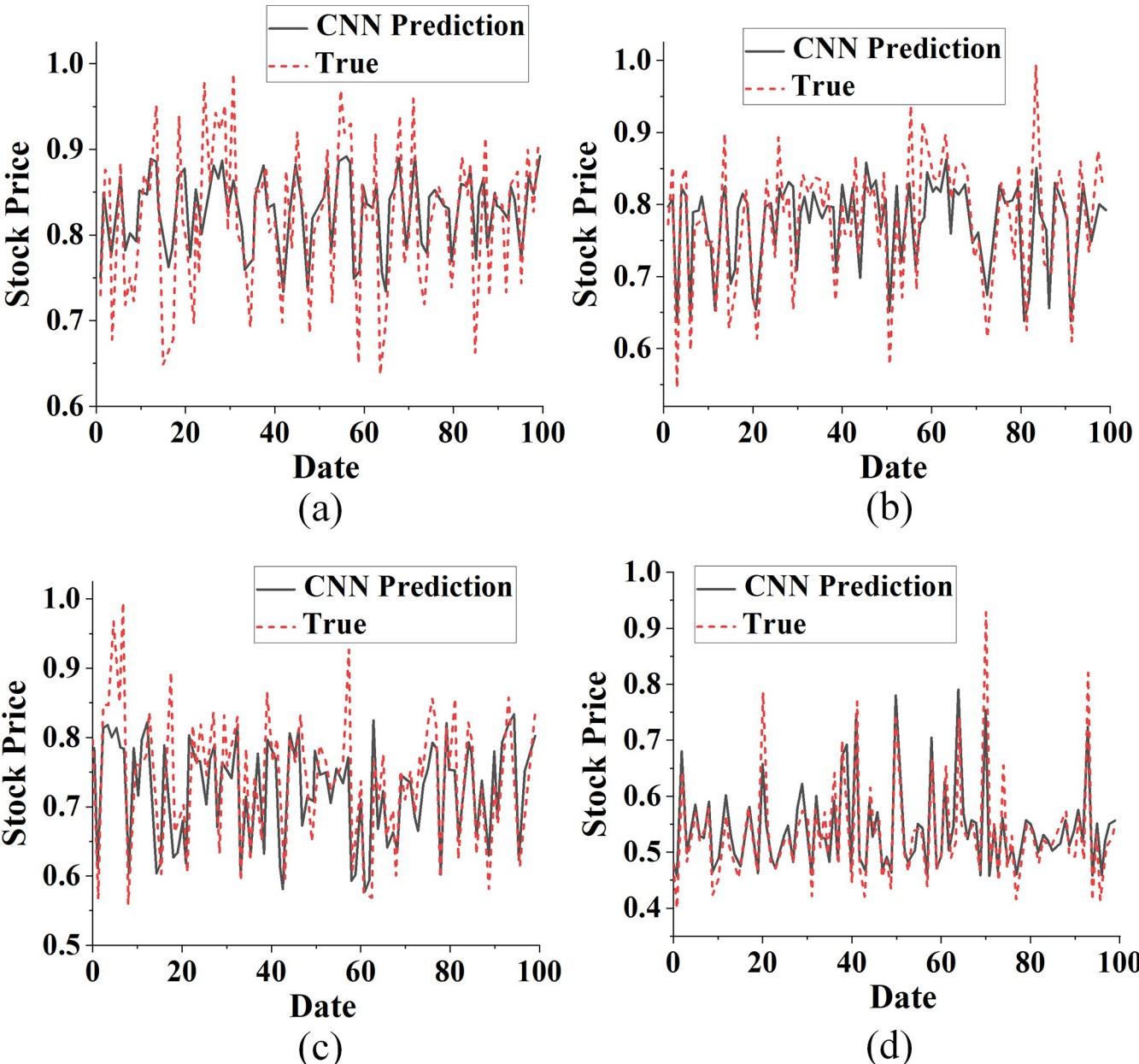

**Fig 6.** The result of predicting the trend of real estate index and stock based on the CNN algorithm compared with the actual value (a. China Real Estate Index (399241); b. Poly Development (600048); c. Gemdale Group (600383); d. Nanshan Holdings (002314).

memory nodes, LSTM can better learn the context between time series and has a natural advantage in stock trend prediction.

Besides, the CNN-LSTM proposed here predicts the trend of real estate index and stock in Fig 8. Under the proposed model, not only the trend of China Real Estate Index (399241), Poly Development (600048), Gemdale Group (600383), and Nanshan Holdings (002314) is effectively predicted, but the predicted value is very similar to the actual value. On the one hand, this may be because CNN's convolutional layer well supplements LSTM in processing input images. On the other hand, LSTM can record historical information and learn the relationship between time series. The CNN-LSTM algorithm consists of two LSTM layers and one

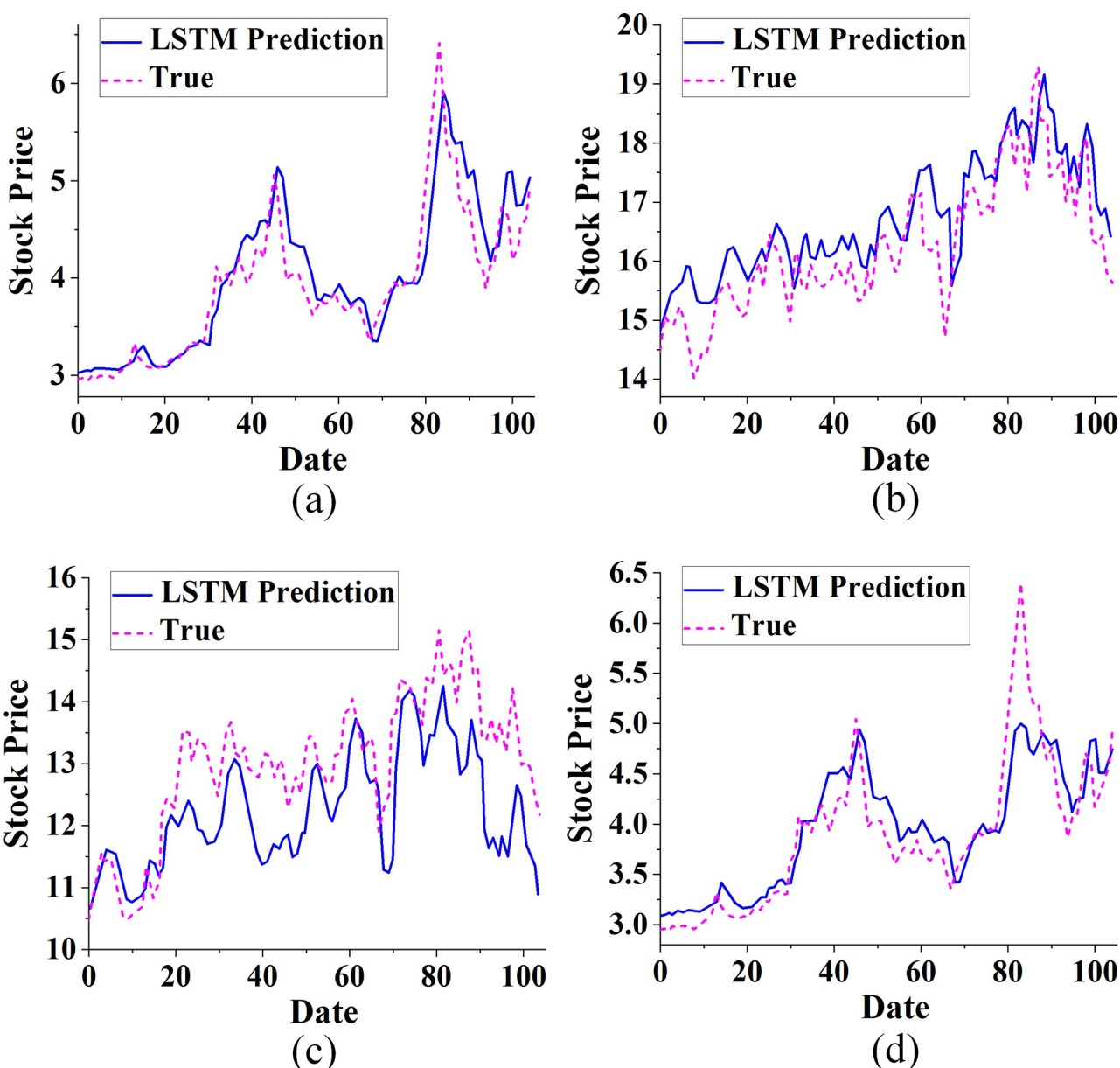

**Fig 7.** The comparison of the trend of real estate index and stock based on the LSTM algorithm with the actual value (a. China Real Estate Index (399,241); b. Poly Development (600048); c. Gemdale Group (600383); d. Nanshan Holdings (002314).

convolution layer, integrating the characteristics of LSTM and CNN and achieving an excellent prediction effect.

Subsequently, the Root Mean Square Error (RMSE) of China Real Estate Index (399241), Poly Development (600048), Gemdi Group (600383), and Nanshan Holdings (002314) under CNN, LSTM, and CNN-LSTM algorithms are calculated. The prediction performance of the three algorithms is judged. RMSE and its average values under CNN, LSTM and CNN-LSTM algorithms are outlined in Table 2.

In Table 2, the purpose is to judge the prediction performance of CNN, LSTM, and CNN-LSTM. Obviously, CNN-LSTM has the highest RMSE, followed by the LSTM algorithm, and the CNN algorithm has the worst prediction performance. This is because two LSTM layers

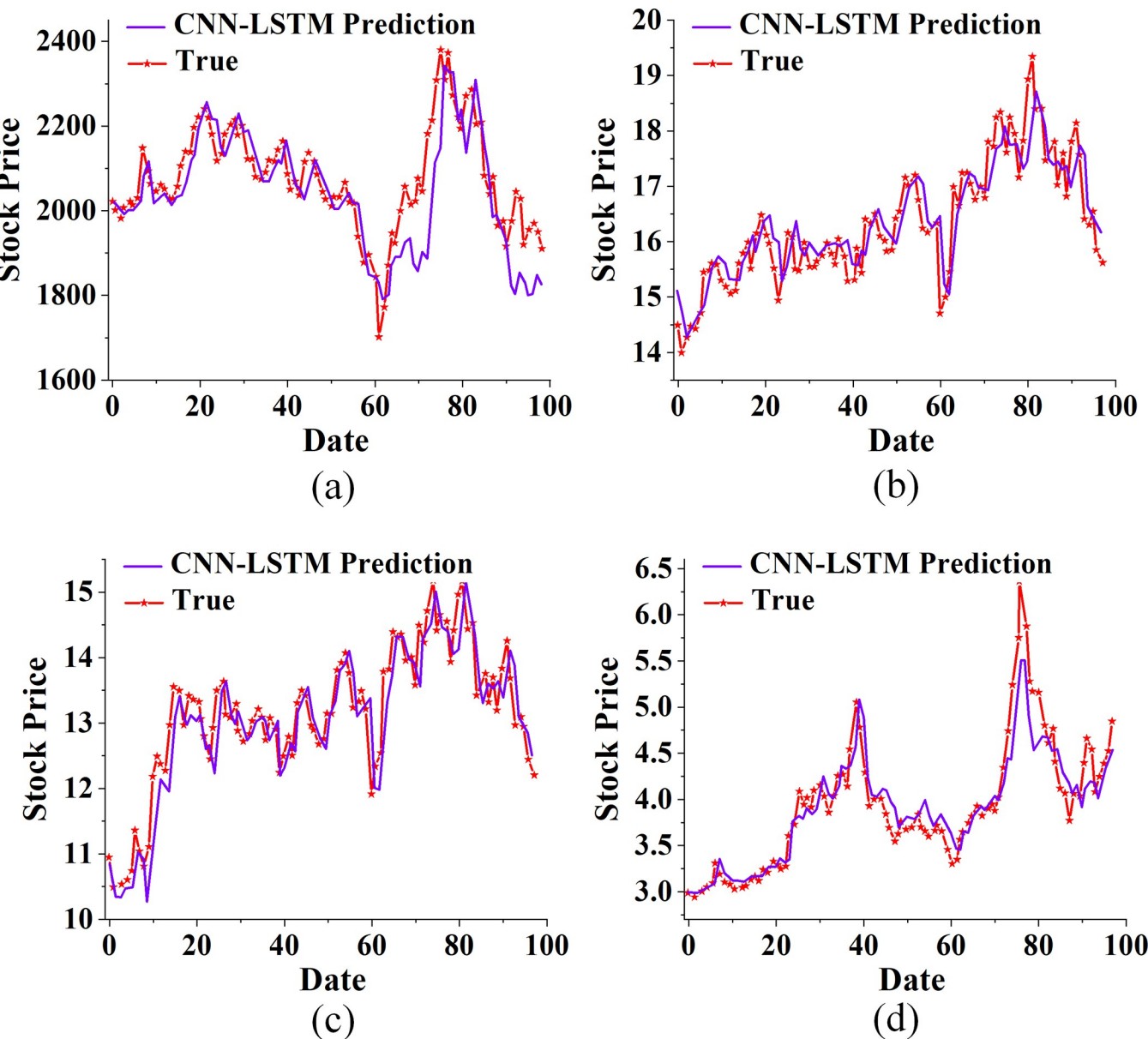

**Fig 8.** The proposed CNN-LSTM algorithm predicting the trend of real estate index and stock (a. China Real Estate Index (399,241); b. Poly Development (600048); c. Gemdale Group (600383); d. Nanshan Holdings (002314).

**Table 2. RMSE and its average values under CNN, LSTM and CNN-LSTM algorithms.**

| | China Real Estate Index (399241) | Poly Development (600048) | Gemdale Group (600383) | Nanshan Holdings. (002314) | The average value of RMSE |
|---|---|---|---|---|---|
| CNN-LSTM | 0.067 | 0.038 | 0.045 | 0.032 | 0.046 |
| LSTM | 69.135 | 0.911 | 1.030 | 0.350 | 17.857 |
| CNN | 86.371 | 0.507 | 0.490 | 0.334 | 21.926 |

**Table 3. The descriptive statistics of the CNN, LSTM, and CNN-LSTM algorithms.**

|  | CNN | LSTM | CNN-LSTM |
|---|---|---|---|
| Number of values | 25 | 25 | 25 |
| Minimum | 0.0004317 | 0.0003458 | 0.0001176 |
| 25% Percentile | 0.0005317 | 0.0005458 | 0.0001776 |
| Median | 0.0005317 | 0.0005458 | 0.0001776 |
| 75% Percentile | 0.0005317 | 0.0005458 | 0.0001776 |
| Maximum | 0.0007317 | 0.0006458 | 0.0002476 |
| Range | 0.00049 | 0.00041 | 0.00009 |
| 10% Percentile | 0.0005117 | 0.0003958 | 0.0001476 |
| 90% Percentile | 0.0007617 | 0.0005858 | 0.0002176 |
| Mean | 0.0005758 | 0.0005558 | 0.0001876 |
| Std. Deviation | 0.0000934 | 0.0000891 | 0.0000235 |
| Std. Error of Mean | 0.0000152 | 0.0000114 | 0.0000071 |
| Sum | 0.04813 | 0.03276 | 0.00875 |

and one convolutional layer are included in the CNN-LSTM. The advantages of LSTM and CNN are combined to predict the real estate index and stock trend. Therefore, the CNN-LSTM algorithm optimized on DL achieves a more accurate real estate index and stock prediction effect.

## 4.2 Statistical analysis

The descriptive statistics of the proposed CNN-LSTM algorithm and other comparison model algorithms are presented in Table 3. The statistical methods of ANOVA and T-test are further applied to compare the algorithms to determine whether there are major differences between the proposed algorithm and other comparison algorithms, as expressed in Tables 4 and 5.

In Table 3, through the descriptive statistics of proposed CNN-LSTM, CNN, and LSTM algorithms, it can be found that the descriptive data of CNN-LSTM algorithm proposed here are the lowest in measurement of concentration tendency, measurement of uncertainty, mean value, median value, and mode. LSTM algorithm is second, and CNN algorithm has the highest value. Therefore, the descriptive statistical analysis in Table 2 shows the superiority of the proposed CNN-LSTM algorithm.

The statistical methods of ANOVA test and T-test are further applied to compare the algorithms, and the results are demonstrated in Tables 4 and 5. It can be found that in the ANOVA test, the proposed CNN-LSTM algorithm has P values less than 0.0001 when compared with CNN and LSTM. In the t test, it is seen that The P value of the proposed CNN-LSTM algorithm compared with CNN and LSTM shows obvious difference (P<0.01). Thus, the ANOVA test and T-test present that the proposed CNN-LSTM algorithm has a better prediction effect.

## 4.3 Comparative analysis of accuracy under different algorithms

The prediction accuracy of CNN algorithm, LSTM algorithm, and the proposed CNN-LSTM algorithm is compared with that of the model algorithm proposed by Kamara (2022) and Song et al. (2023), as suggested in Fig 9.

**Table 4. ANOVA test results of the proposed CNN-LSTM, CNN, and LSTM algorithms.**

|  | SS | DF | MS | F (DFn, DFd) | P value |
|---|---|---|---|---|---|
| Treatment (between columns) | 0.000318 | 5 | 6.51E-05 | F (5, 65) = 176.4 | P<0.0001 |
| Residual (within columns) | 0.000225 | 65 | 3.48E-07 | - | - |
| Total | 0.000543 | 70 | - | - | - |

**Table 5. Descriptive one-sample t test results of the CNN, LSTM, and proposed CNN-LSTM algorithms.**

| | CNN | LSTM | CNN-LSTM |
|---|---|---|---|
| Theoretical mean | 0 | 0 | 0 |
| Actual mean | 0.000427 | 0.000331 | 0.000063 |
| Number of values | 25 | 25 | 25 |
| One sample t test | | | |
| t, df | t = 62.18, df = 24 | t = 47.94, df = 24 | |
| P value (two tailed) | 0.0001 | 0.0001 | |
| P value summary | **** | **** | |
| significant (alpha = 0.05)? | Yes | Yes | |
| How big is the discrepancy? | | | |
| SD of discrepancy | 0.000427 | 0.000331 | |
| SEM of discrepancy | 3.27E-04 | 4.58E-06 | |
| 95% confidence interval | 7.76E-06 | 2.15E-06 | |
| R squared (partial eta squared) | 0.9935 | 09921 | |

Overall, the predicted accuracy of all algorithms shows a rapid increase and then basically remains unchanged until the training is completed on the training set. Among them, the prediction accuracy of this study reaches 90.55%, which is at least 1.74% higher than that of the

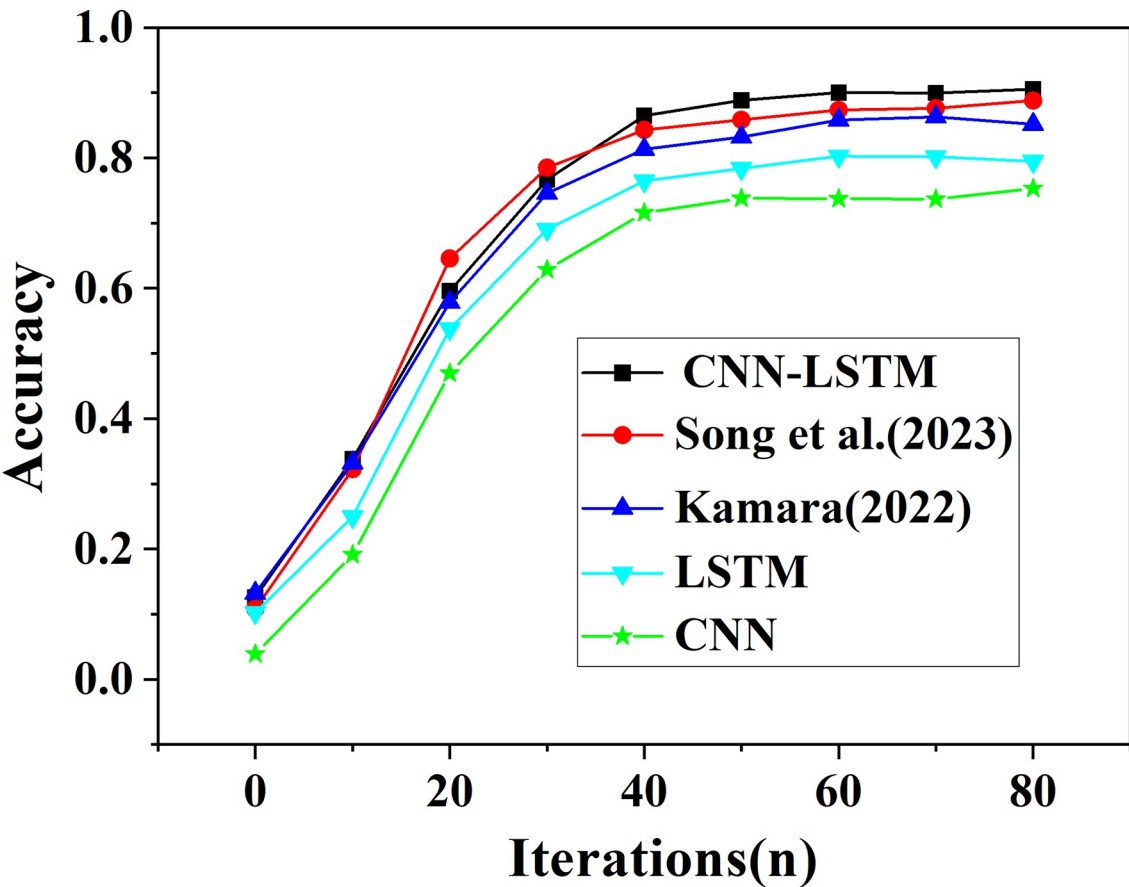

**Fig 9. Influence curve of stock prediction accuracy with increased iterations under different algorithms.**

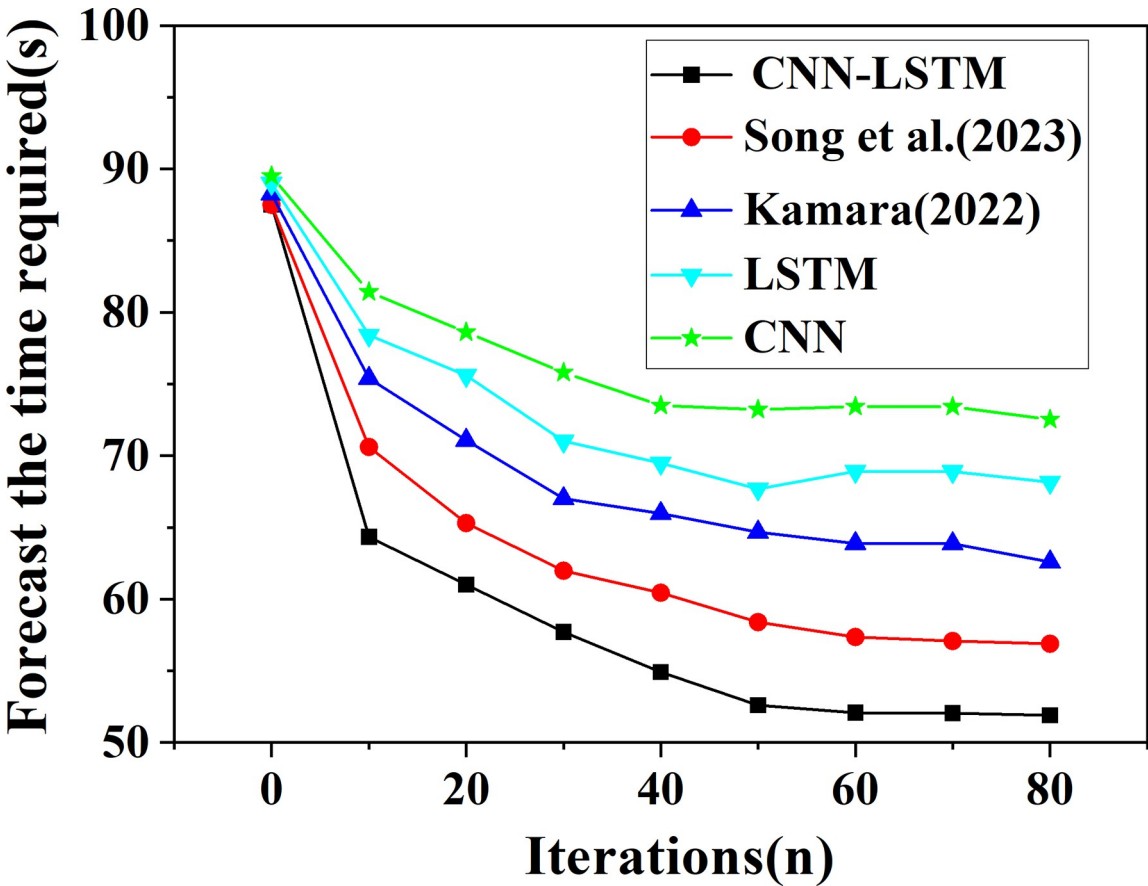

**Fig 10. The influence curve of the time required for stock prediction with the increase of iteration times under different algorithms.**

model algorithm proposed by other scholars. In addition, the prediction accuracy of each algorithm for real estate index and stock trend in descending order is CNN-LSTM optimization algorithm >Song et al. (2023)>Kamara (2022)> LSTM algorithm >CNN algorithm. The prediction accuracy's order is consistent with that of RMSE. Therefore, compared with the model algorithms, the proposed CNN-LSTM algorithm optimized by DL has significantly better prediction accuracy for the real estate index and stock trend prediction.

The proposed CNN-LSTM algorithm is further compared with the time required for prediction by other model algorithms, as portrayed in Fig 10.

The prediction time required by each algorithm is further compared and analyzed, and the results are displayed in Fig 10. It can be found that with the increase of the number of iterations, the required prediction time shows a trend of first increasing and then basically unchanged, that is, convergence is achieved. Compared with other algorithm models, the prediction time required by the CNN-LSTM model algorithm proposed here is basically stable at 52.05s, which is significantly less than the prediction time required by other algorithms. This may be because the proposed CNN parallel optimization processing combined with LSTM algorithm can enhance the generalization ability and accelerate the convergence rate of the model training process. Consequently, the proposed CNN-LSTM optimization algorithm based on DL can achieve higher prediction effect in a shorter period of time for the real estate index and stock trend prediction model algorithm.

**4.4 Discussion**

In this study, the performance of CNN-LSTM algorithm based on DL optimization to real estate index and stock trend prediction model algorithm is analyzed through simulation experiments. Firstly, the CNN, LSTM and the proposed CNN-LSTM algorithms are compared with the real "closing price" data of China Real Estate Index (399241), Poly Development (600048), Gedi Group (600383), and Nanshan Holdings (002314), respectively. It can be found that the proposed CNN-LSTM algorithm can effectively predict each index and the price of stock, and the predicted value is very close to the actual value. This may be because CNN algorithm enables the convolution layer to play a better role, while LSTM can record historical information and learn the relationship between the time series. The constructed CNN-LSTM optimization algorithm includes two LSTM layers and one convolutional layer, which integrates the characteristics of the LSTM model and the CNN model, so as to achieve a more accurate prediction effect. The optimization effect of this algorithm is consistent with that of the algorithm proposed by Abualigah et al. (2021) [29].

The performance of the proposed algorithm is further analyzed from the perspective of statistics. After the application of ANOVA test and T-test, it is obvious that the P values of the proposed CNN-LSTM algorithm compared with CNN and LSTM are less than 0.0001, with significant statistical difference. Thus, the effective prediction advantage of the constructed CNN-LSTM optimization algorithm has its mathematical basis, which is consistent with the views of Ibrahim et al. (2021) [30] and Salamai et al. (2021) [31].

# 5. Conclusion

China's commercial real estate volume and potential are huge. The combination of commercial real estate with tourism, culture, and infrastructure in China plays a crucial role in social and economic development. Firstly, this study introduces and combines CNN and LSTM algorithms in DL to predict the real estate index and stock trend. Secondly, the proposed CNN-LSTM is verified by numerous real estate indexes, such as Poly Development, Gemdale Group, and Nanshan Holdings. Finally, the proposed CNN-LSTM algorithm is more accurate in predicting the trend of the China real estate index and the stocks of real estate companies. It significantly contributes to intelligent prediction of the real estate field trend.

# 6. Limitations and prospects

Of course, there are also some shortcomings in this study. For example, the LSTM-CNN algorithm constructed here only contains two-layer LSTM and one-layer convolution. Although the prediction accuracy of the proposed model is optimal, the multi-layer LSTM or convolutional layer can still be considered to further optimize the network structure in the subsequent work. Moreover, factors such as domestic and foreign policy environment, economic situation, corporate financial statements and executive personnel changes are not taken into account in the model. Meanwhile, if the stock trading is carried out by T+1 strategy, frequent trading will involve high commission fees. These factors may have an impact on the future trend of the stock, and further introducing relevant factors into it is also the direction of subsequent exploration.

# Supporting information

**S1 Data.**
(ZIP)

## Author Contributions

**Data curation:** Ningyan Chen.

**Writing – original draft:** Ningyan Chen.

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
