## [Decision Letter · Decision Letter 0]

16 Jan 2023

PONE-D-22-30740Visual Recognition and Prediction Analysis of China's Real Estate Index and Stock Trend based on CNN-LSTM Algorithm Optimized by Neural NetworksPLOS ONE

Dear Dr. Chen,

Thank you for submitting your manuscript to PLOS ONE. After careful consideration, we feel that it has merit but does not fully meet PLOS ONE’s publication criteria as it currently stands. Therefore, we invite you to submit a revised version of the manuscript that addresses the points raised during the review process.

Please include the following items when submitting your revised manuscript:A rebuttal letter that responds to each point raised by the academic editor and reviewer(s). You should upload this letter as a separate file labeled 'Response to Reviewers'.A marked-up copy of your manuscript that highlights changes made to the original version. You should upload this as a separate file labeled 'Revised Manuscript with Track Changes'.An unmarked version of your revised paper without tracked changes. You should upload this as a separate file labeled 'Manuscript'.

We look forward to receiving your revised manuscript.

Kind regards,

Sathishkumar V E

Academic Editor

PLOS ONE

Journal Requirements:

4. We note you have included a table to which you do not refer in the text of your manuscript. Please ensure that you refer to Table 7 in your text; if accepted, production will need this reference to link the reader to the Table

Reviewers' comments:

Reviewer's Responses to Questions

**Comments to the Author**

1. Is the manuscript technically sound, and do the data support the conclusions?

Reviewer #1: Yes

Reviewer #2: No

2. Has the statistical analysis been performed appropriately and rigorously? 

Reviewer #1: N/A

Reviewer #2: No

3. Have the authors made all data underlying the findings in their manuscript fully available?

Reviewer #1: Yes

Reviewer #2: No

4. Is the manuscript presented in an intelligible fashion and written in standard English?

Reviewer #1: Yes

Reviewer #2: No

5. Review Comments to the Author

Reviewer #1: ......

The language of this paper can be improved to be clearer for the reader. A significant improvement is needed to take another round of consideration.

The authors should improve the abstract.

The abstract should contain the problem, background, proposed method, and result.

Using a better English language, the authors should improve the paper abstract.

An abstract should focus on the problem, state of the art of method, and explain results—no need to explain further research in the abstract.

The motivation of the work is not clear. Add the main motivation of this proposal and what is the main difference between this proposal and existing methods.

Clarify the novelty of the techniques presented and the problem addressed.

The scientific contribution of this work to the state of the art is not clearly stated.

The originality of the contribution, as well as the benefits that it provides to previous ones, must be clear.

The authors did not provide proof of the correctness of their proposed method. I am talking about mathematical proofs.

Some works should mention in this paper:

Aquila Optimizer: A novel meta-heuristic optimization algorithm

The arithmetic optimization algorithm

Advances in Sine Cosine Algorithm: A comprehensive survey

Feature selection and enhanced krill herd algorithm for text document clustering

comparisons with other state-of-the-art methods should be added.

A clear justification for the proposed method compared to other current methods shows a clear motivation for this research.

The figures quality are very week. It should be improved.

few references and most of them are old. new high-quality references should be added.

Add the time complexity of the proposed method and compare it with the original method.

What is the main motivation behind this proposal?

The tables are not clear in terms of presentations.

Reviewer #2: The title can be improved.

Enhance the abstract and introduction to show the motivation for this work.

A comparative study can be added to the related work section in table form to show the recent efforts.

The authors should provide more details regarding the analysis of the results.

what are the parameters configuration for the proposed Algorithm ?

How to initialize the agents in the proposed Algorithm?

There should be some discussion on the limitations of the methods presented in a separate section.

how to tune LSTM hyperparameters and its value

How to initialize the agents in the proposed Algorithm?

Some additional experiments are required:

a. - Scalability

b. - Runtime

c. - Memory

d. - Sensitivity analysis

Read and cite these references.

Ibrahim, S. Mirjalili, M. El-Said, S. Ghoneim, M. Al-Harthi et al., "Wind speed ensemble forecasting based on deep learning using adaptive dynamic optimization algorithm," IEEE Access, vol. 9, no. 1, pp. 125787-125804, 2021.

It is necessary to discuss the complexity of the proposed Algorithm.

Read and cite these references.

Salamai, E.-S. M. El-kenawy and A. Ibrahim, “Dynamic Voting Classifier for Risk Identification in Supply Chain 4.0,” Computers Materials & Continua, vol. 69, no. 3, pp. 3749-3766, 2021.

Statistical analysis should be carried out to demonstrate that the experimental results are significant. Such as the ANOVA test and T-test

Read and cite these references.

E.-S. M. El-kenawy, H. F. Abutarboush, A. W. Mohamed and A. Ibrahim, “Advance artificial intelligence technique for designing double T-shaped monopole antenna,” Computers Materials & Continua, vol. 69, no. 3, pp. 2983-2995, 2021.

Some syntax errors or improper expressions exist in the manuscript.

More up-to-date studies are suggested to be cited.

6. PLOS authors have the option to publish the peer review history of their article (what does this mean?). If published, this will include your full peer review and any attached files.

Reviewer #1: No

Reviewer #2: No

<quillbot-extension-portal></quillbot-extension-portal>

---

## [Author Response · Author response to Decision Letter 0]

1 Feb 2023

4. We note you have included a table to which you do not refer in the text of your manuscript. Please ensure that you refer to Table 7 in your text; if accepted, production will need this reference to link the reader to the Table

Reply: Thanks for the suggestion. Table 7 that appeared in the manuscript has been revised to Table 2, and it has been cited at the same time, that is, the paragraph above Table 2 in Section 4.1 has been quoted accordingly.

Reviewers' comments:

Reviewer's Responses to Questions

Comments to the Author

1. Is the manuscript technically sound, and do the data support the conclusions?

Reviewer #1: Yes

Reviewer #2: No

Reply: Thanks for the suggestion. The data required for this research have been provided in Section 3.4 of the manuscript, namely "The date, opening, high, close, low, and turnover of Sohu Securities China Real Estate Index (399241), Poly Development (600048), Gedi Group (600383), and Nanshan Holdings (002314) are collected by tushare for the experiment. The data cover November 12, 2019, to May 12, 2022.”. And Sections 4.1-4.3 of the results part are all obtained based on this data. In order to reflect the credibility of the obtained results, the statistical analysis using the ANOVA test and T test in Section 4.2 and the discussion related content in Section 4.4 have been added.

2. Has the statistical analysis been performed appropriately and rigorously?

Reviewer #1: N/A

Reviewer #2: No

Reply: Thanks for the suggestion. The statistical analysis has been carried out in Section 4.2 of the manuscript, and the ANOVA test and T test have been applied to prove that the advantages of the proposed CNN-LSTM algorithm are significant.

3. Have the authors made all data underlying the findings in their manuscript fully available?

Reviewer #1: Yes

Reviewer #2: No

Reply: Thanks for the suggestion. All the data in the results section have been provided, and the TIF format of all the figures has been provided as requested.

4. Is the manuscript presented in an intelligible fashion and written in standard English?

Reviewer #1: Yes

Reviewer #2: No

Reply: Thanks for the suggestion. The grammatical errors and wording problems in the manuscript have been revised to further enhance the readability of the manuscript.

5. Review Comments to the Author

Reviewer #1: ......

The language of this paper can be improved to be clearer for the reader. A significant improvement is needed to take another round of consideration.

Reply: Thanks for the suggestion. The grammatical errors and wording problems in the manuscript have been revised to further enhance the readability of the manuscript.

The authors should improve the abstract.

The abstract should contain the problem, background, proposed method, and result.

Reply: Thanks for the suggestion. The abstract section has been made to contain various sections such as background, motivation or problem, methods, results, conclusions, etc. Among them, the background is that with the rapid development of Internet technology today, the changing trend of real estate finance has brought a great impact on the development of social economy; the motivation or problem is to explore the visual recognition effect of deep learning on real estate stock trends; the method is to construct The CNN-LSTM algorithm based on neural network optimization used to visually identify and predict the real estate index and stock trends; the result is that the prediction accuracy of the CNN-LSTM optimization algorithm is 90.55%, and the time required for prediction is only 52.05s, and the method verifies the advantages of the algorithm; the conclusion is that the algorithm has a more accurate prediction effect.

Using a better English language, the authors should improve the paper abstract.

Reply: Thanks for the suggestion. The abstract has been rewritten to include background, motivation or problems, methods, results, conclusions, etc., and grammar and wording problems no longer exist, thereby enhancing its readability.

An abstract should focus on the problem, state of the art of method, and explain results—no need to explain further research in the abstract.

Reply: Thanks for the suggestion. The problem, the status of the method, and the results have been described in the abstract. Among them, the problem is to explore the effect of deep learning on the visual recognition of real estate stock trends; the method is to construct a CNN-LSTM algorithm based on neural network optimization for visual recognition and prediction models of real estate indexes and stock trends; the result is the prediction accuracy of the CNN-LSTM optimization algorithm is 90.55%, the time required for prediction is only 52.05s, and the advantages of the algorithm are verified by statistical methods; the conclusion is that the algorithm has a more accurate prediction effect.

The motivation of the work is not clear. Add the main motivation of this proposal and what is the main difference between this proposal and existing methods.

Reply: Thanks for the careful reading. The motivation for this research has been explained at the beginning of the abstract of the manuscript, that is, to explore the visual recognition effect of the CNN-LSTM algorithm based on neural network optimization on China's real estate index and stock trends. At the same time, the difference between the research method and the existing method is added in the relevant content of the abstract results, reflecting the prediction advantages of the CNN-LSTM optimization algorithm and the short prediction time required.

Clarify the novelty of the techniques presented and the problem addressed.

Reply: Thanks for the suggestion. The novelty of this research has been explained in the last paragraph of the introduction of the manuscript, that is, aiming at the openness of the real estate finance field, by improving the deep learning algorithm, the CNN-LSTM algorithm based on deep learning optimization was finally constructed. At the same time, the problems and motivations to be solved in this study are also explained in this paragraph and the abstract, that is, the real estate index and stock trends are affected by many factors and are very open.

The scientific contribution of this work to the state of the art is not clearly stated.

Reply: Thanks for the suggestion. The contribution of this research has been elaborated in the last paragraph of the introduction of the manuscript, which is to promote the intelligent prediction of stock trends and the optimization of corporate structure in the field of real estate finance.

The originality of the contribution, as well as the benefits that it provides to previous ones, must be clear.

Reply: Thanks for the careful reading. The novelty of this study has been explained in the last paragraph of the introduction of the manuscript, that is, in view of the openness of the real estate finance field, CNN is rarely used in financial forecasting. This study improved the deep learning algorithm. Finally, a CNN-LSTM algorithm based on deep learning optimization was constructed to predict the real estate index and stock trends. At the same time, the contribution was explained in this paragraph, which is to promote the intelligent forecasting of stock trends in the field of real estate finance and the optimization of the company structure.

The authors did not provide proof of the correctness of their proposed method. I am talking about mathematical proofs.

Reply: Thanks for the suggestion. The statistical analysis has been added to the results and discussion section, and the CNN-LSTM algorithm proposed in this research has a better prediction effect by applying the statistical methods of ANOVA test and t-test.

Some works should mention in this paper:

Aquila Optimizer: A novel meta-heuristic optimization algorithm

The arithmetic optimization algorithm

Advances in Sine Cosine Algorithm: A comprehensive survey

Feature selection and enhanced krill herd algorithm for text document clustering

Reply: Thanks for the careful reading. The above-mentioned studies have been cited in the manuscript, respectively [10], [16], and [29]. Among them, the content of "Feature selection and enhanced krill herd algorithm for text document clustering" is not consistent with this study, so it is not cited.

comparisons with other state-of-the-art methods should be added.

Reply: Thanks for the suggestion. The comparison between the algorithm proposed in this study and the most advanced algorithm has been added, as mentioned in Figure 9, compared with the model algorithm proposed by scholars in related fields, Kamara (2022) and Song et al. (2023).

A clear justification for the proposed method compared to other current methods shows a clear motivation for this research.

Reply: Thanks for the careful reading. The CNN-SLTM algorithm proposed in this research has been compared with the existing algorithms CNN and SLTM algorithms in Figure 9, and the model proposed by scholars in related fields, Kamara (2022) and Song et al. (2023) for comparison. At the same time, in the discussion part, the obtained results are clearly analyzed and discussed, which reflects the advantage of the higher prediction accuracy of this study.

The figures quality are very week. It should be improved.

Reply: Thanks for the careful reading. The pictures in the manuscript have been optimized and improved. For example, the colors of the arrows in Figure 1, Figure 4, and Figure 5 have been unified, the shadows in Figures 1-5 have been removed, and the display form of Figure 4 has been optimized, etc.

few references and most of them are old. new high-quality references should be added.

Reply: Thanks for the suggestion. The number of documents has been added, making it more than 30. At the same time, more high-quality new documents have been added, respectively [1-4], [8-10], [15,16], and [28 -31].

Add the time complexity of the proposed method and compare it with the original method.

Reply: Thanks for the careful reading. The CNN-SLTM algorithm proposed in this research and the existing algorithm CNN, SLTM algorithm have been added in Section 4.3 of the manuscript. The comparison results of the required time are shown in Figure 10. It can be found that the CNN-SLTM algorithm proposed in this study can achieve higher prediction results in a shorter time.

What is the main motivation behind this proposal?

Reply: Thanks for the suggestion. The motivation for this study has been explained in the last paragraph of the introduction and the abstract of the manuscript, that is, to explore the vision of China's real estate index and stock trends based on the CNN-LSTM algorithm optimized by neural network in view of the openness of the real estate financial field. The recognition effect finally makes the real estate company structure more optimized.

The tables are not clear in terms of presentations.

Reply: Thanks for the careful reading. Tables 2-5 appearing in the manuscript have been described in detail, namely the RMSE value results of CNN, LSTM and CNN-LSTM algorithms, descriptive statistics results, ANOVA test results and t-test results. This further enhances the readability of this study.

Reviewer #2: The title can be improved.

Enhance the abstract and introduction to show the motivation for this work.

Reply: Thanks for the careful reading. The abstract has been rewritten to show the motivation or problem of this research, that is, to explore the effect of deep learning on the visual recognition of real estate stock trends, in order to provide information for the subsequent intelligent prediction of real estate indexes and real estate trends and the structural optimization of real estate companies. At the same time, the introduction part of the manuscript has been optimized. First, the background and motivation of this research are explained in the first paragraph, that is, the rapid growth of the real estate industry makes the subprime mortgage crisis very easy to occur, so it is very important to accurately predict its trend. In the second paragraph, the algorithms involved are described, such as CNN, RNN, etc. In the last paragraph, the significance, innovations, and contributions of this research are described, that is, the CNN-LSTM algorithm based on deep learning optimization is constructed. The aim is to provide a reference for the follow-up real estate index and real estate trend intelligent prediction and the optimization of the company structure.

A comparative study can be added to the related work section in table form to show the recent efforts.

Reply: Thanks for the suggestion. First of all, the literature in Section 2 has been updated, namely references [10] and [15,16], so as to display the latest research in related fields. Second, in Section 2.3, the advantages and disadvantages of related research are analyzed. Although many scholars try to predict the financial trends of their indexes and stocks, there are still great unknowns and research gaps in the changes in the real estate industry, which highlights the advantages of this research, that is, the application is relatively large. A few CNNs are introduced to optimize the algorithm model, and an exploration is made on the prediction of indexes and stocks in the real estate field.

The authors should provide more details regarding the analysis of the results.

Reply: Thanks for the suggestion. The statistical analysis section has been added to the results and discussion section of the manuscript, and the advantages of the algorithm proposed in this study are reflected by applying the statistical methods of ANOVA and t-test. At the same time, a discussion section is added to discuss and analyze the results of this research in more detail, and finally highlight the results of this research algorithm.

what are the parameters configuration for the proposed Algorithm ?

Reply: Thanks for the suggestion. The CNN-LSTM algorithm proposed in this study and the parameter configuration of CNN and LSTM have been described in Section 3.4 of the manuscript, as shown in Table 1, including the CNN-LSTM algorithm and each neural network layer of CNN and LSTM and output dimension values.

How to initialize the agents in the proposed Algorithm?

Reply: In Section 3.4 of the manuscript, the relevant initialization values and dimensions of the input layer of the CNN-LSTM algorithm proposed in this research have been described in detail, namely "(batch, 30, 6)".

There should be some discussion on the limitations of the methods presented in a separate section.

Reply: Thanks for the suggestion. The limitations and prospects of this research have been discussed in Section 6 of the manuscript. For example, the LSTM-CNN algorithm only includes two layers of LSTM layers and one layer of convolutional layers. Consider using more convolutional layers or LSTM layers to optimize the network, etc.

how to tune LSTM hyperparameters and its value

Reply: Thanks for the careful reading. The hyperparameters of the LSTM algorithm have been set in Section 3.4 of the manuscript, which includes the adjustment of the output dimension value of the input layer, lstm1, lstm2 and output layer.

How to initialize the agents in the proposed Algorithm?

Reply: Thanks for the suggestion. In Section 3.4 of the manuscript, the relevant initialization values and dimensions of the input layer of the CNN-LSTM algorithm proposed in this research have been described in detail, namely "(batch, 30, 6)".

Some additional experiments are required:

a. - Scalability

b. - Runtime

c. - Memory

d. - Sensitivity analysis

Read and cite these references.

Ibrahim, S. Mirjalili, M. El-Said, S. Ghoneim, M. Al-Harthi et al., "Wind speed ensemble forecasting based on deep learning using adaptive dynamic optimization algorithm," IEEE Access, vol. 9, no. 1, pp. 125787-125804, 2021.

Reply: Thanks for the suggestion. Additional experiments have been carried out in the manuscript, as shown in Table 3-Table 5, the ANOVA test and T test in the application of statistical methods are shown to reflect the significance of the prediction effect of this study; the figure 10 is added. The running time required for the prediction of each algorithm is used to demonstrate the complexity of the model algorithm proposed in this study. After discussion, it is considered that these experimental results are sufficient to reflect the advantages of this study. At the same time, the literature was cited in the manuscript as reference [30].

It is necessary to discuss the complexity of the proposed Algorithm.

Read and cite these references.

Salamai, E.-S. M. El-kenawy and A. Ibrahim, “Dynamic Voting Classifier for Risk Identification in Supply Chain 4.0,” Computers Materials & Continua, vol. 69, no. 3, pp. 3749-3766, 2021.

Reply: Thanks for the suggestion. In order to analyze the complexity of the model constructed in this research, a comparative experiment was carried out by introducing the required running time of each algorithm to predict, as shown in Figure 10, that is, the shorter the running time, the lower the complexity of the algorithm. At the same time, the literature was cited in the manuscript as reference [31].

Statistical analysis should be carried out to demonstrate that the experimental results are significant. Such as the ANOVA test and T-test

Read and cite these references.

E.-S. M. El-kenawy, H. F. Abutarboush, A. W. Mohamed and A. Ibrahim, “Advance artificial intelligence technique for designing double T-shaped monopole antenna,” Computers Materials & Continua, vol. 69, no. 3, pp. 2983-2995, 2021.

Reply: Thanks for the suggestion. Statistical analysis has been carried out in the manuscript, and ANOVA test and T test have been applied to prove that the advantages of the proposed CNN-LSTM algorithm are significant. At the same time, the literature was cited in the manuscript as reference [28].

Some syntax errors or improper expressions exist in the manuscript.

Reply: Thanks for the suggestion. The grammatical errors and wording problems in the manuscript have been revised to further enhance the readability of the manuscript. 

More up-to-date studies are suggested to be cited. 

Reply: Thanks for the suggestion. More recent literature studies have been cited as [1-4], [8-10], [15,16], and [28-31].

---

## [Decision Letter · Decision Letter 1]

9 Feb 2023

Visual Recognition and Prediction Analysis of China's Real Estate Index and Stock Trend based on CNN-LSTM Algorithm Optimized by Neural Networks

PONE-D-22-30740R1

Dear Dr. Chen,

We’re pleased to inform you that your manuscript has been judged scientifically suitable for publication and will be formally accepted for publication once it meets all outstanding technical requirements.

Kind regards,

Sathishkumar V E

Academic Editor

PLOS ONE

Additional Editor Comments (optional):

Reviewers' comments:

Reviewer's Responses to Questions

**Comments to the Author**

1. If the authors have adequately addressed your comments raised in a previous round of review and you feel that this manuscript is now acceptable for publication, you may indicate that here to bypass the “Comments to the Author” section, enter your conflict of interest statement in the “Confidential to Editor” section, and submit your "Accept" recommendation.

Reviewer #1: All comments have been addressed

2. Is the manuscript technically sound, and do the data support the conclusions?

Reviewer #1: Yes

3. Has the statistical analysis been performed appropriately and rigorously? 

Reviewer #1: Yes

4. Have the authors made all data underlying the findings in their manuscript fully available?

Reviewer #1: Yes

5. Is the manuscript presented in an intelligible fashion and written in standard English?

Reviewer #1: Yes

6. Review Comments to the Author

Reviewer #1: (No Response)

7. PLOS authors have the option to publish the peer review history of their article (what does this mean?). If published, this will include your full peer review and any attached files.

Reviewer #1: No

<quillbot-extension-portal></quillbot-extension-portal>

---

## [Editor Report · Acceptance letter]

15 Feb 2023

PONE-D-22-30740R1 

Visual Recognition and Prediction Analysis of China's Real Estate Index and Stock Trend based on CNN-LSTM Algorithm Optimized by Neural Networks 

Dear Dr. Chen:

I'm pleased to inform you that your manuscript has been deemed suitable for publication in PLOS ONE. Congratulations! Your manuscript is now with our production department. 

Kind regards, 

on behalf of

Dr. Sathishkumar V E 

Academic Editor

PLOS ONE